# Computing the Atom Graph of a Graph and the Union Join Graph of a Hypergraph

**Anne Berry** [1] **and Geneviève Simonet** [2],*

1  LIMOS UMR CNRS 6158, Ensemble Scientifique des Cézeaux, F-63173 Aubière, France; berry@isima.fr
2  LIRMM, 161 Rue Ada, F-34392 Montpellier, France
*  Correspondence: genevieve.simonet@umontpellier.fr

**Abstract:** The atom graph of a graph is a graph whose vertices are the atoms obtained by clique minimal separator decomposition of this graph, and whose edges are the edges of all possible atom trees of this graph. We provide two efficient algorithms for computing this atom graph, with a complexity in $O(min(n^\omega \log n, nm, n(n + \overline{m})))$ time, where $n$ is the number of vertices of $G$, $m$ is the number of its edges, $\overline{m}$ is the number of edges of the complement of $G$, and $\omega$, also denoted by $\alpha$ in the literature, is a real number, such that $O(n^\omega)$ is the best known time complexity for matrix multiplication, whose current value is 2,3728596. This time complexity is no more than the time complexity of computing the atoms in the general case. We extend our results to $\alpha$-acyclic hypergraphs, which are hypergraphs having at least one join tree, a join tree of an hypergraph being defined by its hyperedges in the same way as an atom tree of a graph is defined by its atoms. We introduce the notion of union join graph, which is the union of all possible join trees; we apply our algorithms for atom graphs to efficiently compute union join graphs.

**Keywords:** clique separator decomposition; atom tree; atom graph; clique tree; clique graph; $\alpha$-acyclic hypergraph





## 1. Introduction

Decomposition by clique minimal separators (into subgraphs called *atoms*) was introduced by Tarjan [1] in 1985 as a useful hole- and antihole-preserving decomposition. It turns out that this decomposition is unique when clique *minimal* separators are used [2].

This decomposition has given rise to recent interest, both in the general case [2–5] and for special graph classes [6–11]. Applications have arisen in the fields of databases [12], text mining [13], and biology [14,15].

Berry et al. [4] introduced the concept of *atom tree*, which organizes the atoms of the clique minimal separator decomposition into a tree as a generalization of the clique tree for chordal graphs: the nodes are the atoms, and the edges correspond to the clique minimal separators of the graph. However, as is the case for the clique tree, the atom tree is not uniquely defined. This can be a problem, for instance, with the promising use of an atom tree as a visualization tool.

In this paper, we focus on the *atom graph*, whose vertices are the atoms and whose edges are those of all possible atom trees.

The notion of atom graph was introduced in 2007 in [15], in the context of visualizing biological clusters. An efficient construction algorithm was proposed in 2010 in [16].

In the case of chordal graphs, atoms are the maximal cliques, and atom trees are clique trees. A related graph that has been extensively studied in this context is the *clique graph* (see, e.g., [17,18]), which is the intersection graph of the maximal cliques. The *weighted clique graph* of a chordal graph has been used to construct a clique tree of this graph [19,20], its clique trees being the maximum weight spanning trees of its weighted clique graph. Thus, the atom graph of a chordal graph is a subgraph of its clique graph. In the context of efficiently constructing a clique tree, in 1991, Blair and Peyton [19] studied the family

of all possible clique trees, an object very close to the atom graph of a chordal graph. In 1995, Galinier et al. [21] used the weighted atom graph of a chordal graph but misguidedly called this object the 'clique graph'. In 2012 in [22], this object is further studied and called the 'reduced clique graph'.

Our first goal in this paper is to propose efficient algorithms to compute the atom graph, both in the general case and in the case of chordal graphs.

Given a graph, all known algorithms for computing the decomposition into atoms first compute a minimal triangulation of the graph [2–4], with the exception of some special graph classes [6,7]. A minimal triangulation can be computed in $O(min(n^\omega \log n, nm, n(n + \overline{m})))$ time, where $n$ is the number of vertices of $G$, $m$ is the number of its edges, $\overline{m}$ is the number of edges of the complement of $G$, and $\omega$, also denoted by $\alpha$ in the literature, is a real number such that $O(n^\omega)$ is the best known time complexity for matrix multiplication, whose current value is 2,3728596 [4,23,24]. From this minimal triangulation, an atom tree can be computed in $O(min(n^\omega, nm, n(n + t)))$ time [2,4,5], where $t$ is the number of two pairs of the minimal triangulation, and thus $t \leq \overline{m}$. As a result, an atom tree can be computed in $O(min(n^\omega \log n, nm, n(n + \overline{m})))$ time.

To compute the atom graph efficiently, we present two different approaches. One takes as input an atom tree as well as the inclusion relation between the separators represented by its edges, and the other takes as input the weighted intersection graph of the atoms. In both cases, we provide an $O(n^2)$ algorithm to compute the atom graph from the input. Our global complexity when taking the graph itself as input comes to $O(min(n^\omega \log n, nm, n(n + \overline{m})))$ time.

We then go on to remark that the atoms of a graph $G = (V, E)$ can be seen as the hyperedges of an $\alpha$-acyclic hypergraph, whose vertex set is $V$, since $G$ has an atom tree, which is a join tree of this hypergraph. However, the atoms of a graph are pairwise non-inclusive, which is not a requirement for $\alpha$-acyclic hypergraphs, where a hyperedge can be included in another. Fortunately, our algorithms also work in this more general context.

We introduce the notion of *union join graph*, which is the union of all join trees, and provide algorithms to compute this object efficiently.

The paper is organized as follows: Section 2 provides some necessary preliminaries. Section 3 discusses useful properties of the atom graph. Section 4 presents our algorithms to compute the atom graph. Section 5 defines the atom hypergraph and relates it to $\alpha$-acyclic hypergraphs. Section 6 discusses how to compute the union join graph of an $\alpha$-acyclic hypergraph. We conclude in Section 7.

## 2. Preliminaries

The graphs considered in this paper are finite and undirected. For a graph $G = (V, E)$, $n = |V|$ and $m = |E|$. For any subset $X$ of $V$, $G(X)$ denotes the subgraph of $G$ induced by $X$. For any vertex $v$ of $G$, $N_G(v)$ denotes the neighborhood of $v$ in $G$: $N_G(v) = \{w \in V \mid vw \in E\}$. For any subset $X$ of $V$, $N_G(X)$ denotes the neighborhood of $X$ in $G$: $N_G(X) = (\cup_{v \in X} N_G(v)) \setminus X$. We will omit the subscripts when there is no ambiguity. A *clique* of $G$ is a set of pairwise adjacent vertices of $G$, and $G$ is *complete* if $V$ is a clique of $G$. The *union of two graphs* $G_1 = (V, E_1)$ and $G_2 = (V, E_2)$ is the graph $G_1 \cup G_2 = (V, E_1 \cup E_2)$.

$\overline{G}$ denotes the *complement* of $G$, and $\overline{m}$ denotes the number of its edges. $\omega$ is a real number, such that $O(n^\omega)$ is the best known time complexity for matrix multiplication. For any set $V$, $\mathcal{P}(V)$ is the power set of $V$. For any subset $\mathcal{A}$ of $\mathcal{P}(V)$, the *intersection graph* of $\mathcal{A}$ is the graph $(\mathcal{A}, E)$, where $E$ is the set of pairs of $\mathcal{A}$ whose intersection is non-empty. For each graph $G$, $\mathcal{K}(G)$ denotes the set of maximal cliques of $G$, and the *clique graph* of $G$ is the intersection graph of $\mathcal{K}(G)$. If $X$ and $Y$ are nodes of a tree $T$, $P_T(X, Y)$ denotes the path in $T$ between $X$ and $Y$.

***Separation.*** Let $S$ be a subset of vertices of a connected graph $G = (V, E)$. $S$ is a *separator* of $G$ if $G(V \setminus S)$ is disconnected. For any vertices $a$ and $b$ in $V \setminus S$, $S$ is an *ab-separator* of $G$ if $a$ and $b$ are in different connected components of $G(V \setminus S)$. $S$ is a *minimal ab-separator* if it is an inclusion-minimal *ab*-separator, and a *minimal separator* if there is some pair $\{a, b\}$

of vertices, such that $S$ is a minimal $ab$-separator. Given a minimal separator $S$, $C$ is a *full component* of $S$ if $C$ is a connected component of $G(V \setminus S)$ and $N_G(C) = S$. $S$ is a minimal separator if and only if $S$ has at least 2 full components, and $S$ is a minimal $ab$-separator if and only if $a$ and $b$ lie in 2 different full components of $S$. Given three subsets $S$, $A$, and $B$ of $V$, $S$ is a (minimal) $AB$-separator of $G$ if it is a (minimal) $ab$-separator of $G$ for each $a \in A$ and each $b \in B$.

A *2-pair* of a connected graph $G$ is a pair $\{x, y\}$ of non-adjacent vertices such that every chordless path between $x$ and $y$ is of length 2, i.e., every sequence of consecutively-adjacent vertices with $x$ and $y$ as endpoints and being minimal for these properties according to the relation of subsequence has 3 vertices, or equivalently, such that $N(x) \cap N(y)$ is a minimal $xy$-separator of $G$. The number of 2-pairs of a graph is denoted $t$, with $t \leq \overline{m}$.

If $G$ is disconnected, then a (minimal) $(ab$-$)$separator of $G$ is a (minimal) $(ab$-$)$separator of one of its connected components. Thus, the set of minimal separators of a graph is the union of the sets of minimal separators of its connected components, and so it is for its set of 2-pairs.

*Chordal graphs.* A graph is *chordal* or *triangulated* if it has no chordless cycle of length at least 4. A graph is chordal if and only if all its minimal separators are cliques [25]. A chordal graph has at most $n$ maximal cliques, and the sum of their sizes is bounded by $n + m$. A connected graph is chordal if and only if it has a *clique tree* [20,26].

**Definition 1.** *Let $G = (V, E)$ be a connected chordal graph. A* clique tree *of $G$ is a tree $T = (\mathcal{K}(G), E_T)$, such that, for each vertex $x$ of $G$, the set $\mathcal{K}_x$ of nodes of $T$ containing $x$ induces a subtree of $T$.*

**Characterization 1** ([19]). *Let $G = (V, E)$ be a connected chordal graph, let $T$ be a clique tree of $G$, and let $S \subseteq V$; then, $S$ is a minimal separator of $G$ if and only if there is an edge $XY$ of $T$, such that $S = X \cap Y$.*

If $G$ is a disconnected chordal graph, we associate with $G$ a forest whose connected components are clique trees of the connected components of $G$. A clique tree (forest) can be computed in linear time [19].

*Atoms.* Atoms are the subgraphs obtained by applying the decomposition by clique minimal separators (see [3] for a survey).

**Characterization 2** ([2]). *An atom of a graph $G = (V, E)$ is an inclusion-maximal subset of $V$ inducing a connected subgraph of $G$ with no clique separator.*

We will denote the set of atoms of $G$ by $\mathcal{A}(G)$.

**Property 1.** *The atoms of a chordal graph are its maximal cliques.*

**Property 2** ([2]). *The intersection of two distinct atoms is a clique.*

**Notation 1.** *For a graph $G = (V, E)$, $G^+$ denotes the graph whose vertex set is $V$ and whose edges are the pairs of $V$ that are contained in a common atom of $G$ (this graph is denoted $G^*$ in [2]).*

**Property 3** ([2]). *For a graph $G$, $G^+$ is chordal, its maximal cliques are the atoms of $G$, and for each clique $S$ of $G$ and each pair $\{a, b\}$ of $V \setminus S$, $S$ is a minimal $ab$-separator of $G$ if and only if $S$ is a minimal $ab$-separator of $G^+$.*

It follows that a graph has at most $n$ atoms.

*Atom trees.* To represent the atoms of a graph, [4] extend the notion of clique tree of a connected chordal graph to the notion of *atom tree* of a connected graph:

**Definition 2** ([4])**.** *Let $G = (V, E)$ be a connected graph. An* atom tree *of G is a tree $T = (\mathcal{A}(G), E_T)$, such that, for each vertex x of G, the set $\mathcal{A}_x$ of nodes of T containing x induces a subtree of T.*

Note that an atom tree is not a decomposition tree of clique separator decomposition as defined in [1,16], though this decomposition is called 'atom tree' in [16].

An atom tree of a connected graph $G$ can be computed in $O(min(n^\omega \log n, nm, n(n + \overline{m})))$ time [4,5].

As for clique trees of chordal graphs, we can extend the definition of an atom tree to the definition of an "atom forest" of an arbitrary graph $G$, whose connected components are atom trees of the connected components of $G$.

The edges of an atom tree (forest) of a graph correspond to its clique minimal separators.

**Characterization 3** ([4])**.** *Let $G = (V, E)$ be a connected graph, let T be an atom tree of G, and let $S \subseteq V$; then S is a clique minimal separator of G if and only if there is an edge AB of T, such that $S = A \cap B$.*

**Property 4.** *For a connected graph G, the atom trees of G are the clique trees of the chordal graph $G^+$ ($G^+$ is defined in Notation 1).*

**Example 1.** *Figure 1 shows a graph G and two of its atom trees. The atoms of G are $A = \{1, 2, 3, 4, 5, 6\}$, $B = \{1, 2, 3, 7\}$, $C = \{1, 7, 8\}$, $D = \{1, 9\}$, $E = \{1, 10, 11\}$, and $F = \{10, 11, 12, 13\}$. Denoting by $\mathcal{A}_x$ the set of atoms containing x for each vertex x of G, the sets $\mathcal{A}_x$ containing at least 2 atoms are $\mathcal{A}_1 = \mathcal{A}(G) \setminus \{F\}$, $\mathcal{A}_2 = \mathcal{A}_3 = \{A, B\}$, $\mathcal{A}_7 = \{B, C\}$, $\mathcal{A}_{10} = \mathcal{A}_{11} = \{E, F\}$. Thus, G has 15 atom trees, which are all the trees obtained from the forest $(\mathcal{A}(G), \{AB, BC, EF\})$ by adding 2 edges not containing the node F (6 containing the edge DE, as the atom tree shown on the left, and 9 not containing it, as the atom tree shown on the right). Each edge XY of each atom tree is labeled with the associated clique minimal separator $X \cap Y$ of G.*

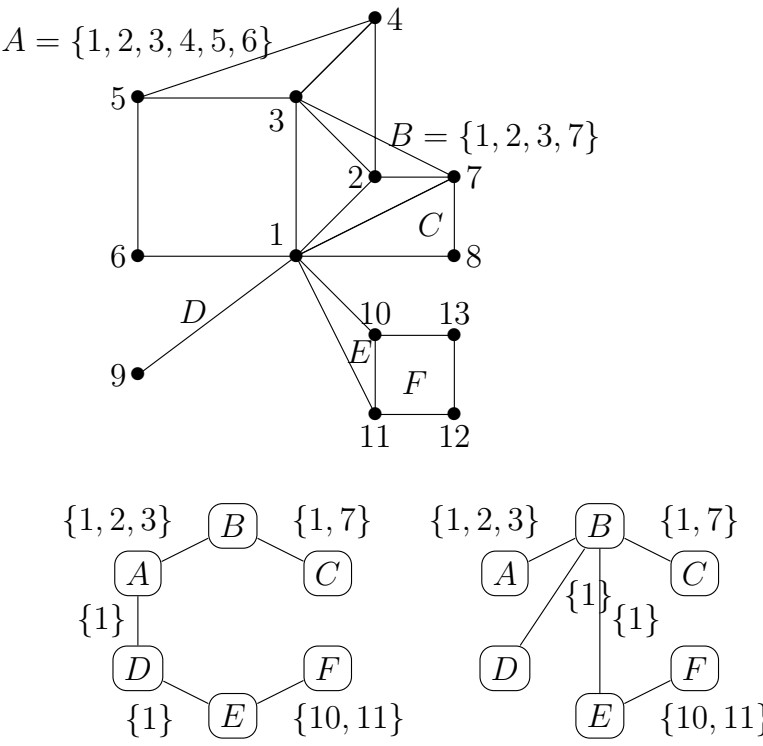

**Figure 1.** A graph $G$ and two atom trees of $G$.

The following properties will be used to compute complexity bounds.

**Property 5.** *Let A and B be distinct atoms of a graph G. Then, $G(A \setminus B)$ is connected and $A \cap B \subseteq N_G(A \setminus B)$.*

**Proof.** By Property 2, $A \cap B$ is a clique of $G$. $A \setminus B$ is connected since otherwise $A \cap B$ would be a clique separator of $G(A)$. Similarly, $A \cap B \subseteq N_G(A \setminus B)$, since otherwise, $A \cap N_G(A \setminus B)$ would be a clique $ab$-separator of $G(A)$ for any $a$ in $A \setminus B$ (which is non-empty by definition of atoms) and any $b$ in $(A \cap B) \setminus N_G(A \setminus B)$. □

**Property 6.** *The sum of the sizes of the atoms of a graph is bounded by $n + m$.*

**Proof.** It is sufficient to prove it in the case of a connected graph $G$. Let $T$ be an atom tree of $G$, and let us show by induction on $|\mathcal{A}|$ that, for each connected subset $\mathcal{A}$ of nodes of $T$, $\Sigma_{X \in \mathcal{A}} |X| \leq n_{\mathcal{A}} + m_{\mathcal{A}}$, where $n\mathcal{A}$ and $m\mathcal{A}$ are the numbers of vertices and edges of the subgraph of $G$ induced by $V_{\mathcal{A}} = \cup_{X \in \mathcal{A}} X$. It trivially holds if $|\mathcal{A}| = 1$. We assume that it holds if $|\mathcal{A}| = k$. Let us show that it holds if $|\mathcal{A}| = k + 1$. Let $X_1$ be a leaf of $T(\mathcal{A})$, let $X_2$ be the neighbor of $X_1$ in $T(\mathcal{A})$, let $S = X_1 \cap X_2$, and let $\mathcal{A}_2 = \mathcal{A} \setminus \{X_1\}$. By induction, hypothesis $\Sigma_{X \in \mathcal{A}_2} |X| \leq n_{\mathcal{A}_2} + m_{\mathcal{A}_2}$. As $T$ is an atom tree of $G$, $V_{\mathcal{A}}$ is the disjoint union of $X_1 \setminus S$ and $V_{\mathcal{A}_2}$, so $n_{\mathcal{A}} = |X_1 \setminus S| + n_{\mathcal{A}_2}$. By Property 5, $S \subseteq N_G(X_1 \setminus S)$, so $m_{\mathcal{A}} \geq |S| + m_{\mathcal{A}_2}$. It follows that $\Sigma_{X \in \mathcal{A}} |X| = |X_1| + \Sigma_{X \in \mathcal{A}_2} |X| \leq (|X_1 \setminus S| + |S|) + (n_{\mathcal{A}_2} + m_{\mathcal{A}_2}) = (|X_1 \setminus S| + n_{\mathcal{A}_2}) + (|S| + m_{\mathcal{A}_2}) \leq n_{\mathcal{A}} + m_{\mathcal{A}}$. □

**Property 7.** *The sum of the sizes of the sets $X \cap Y$ for each edge $XY$ of an atom tree $T$ of a connected graph is bounded by $n + m$, and these sets can be computed from T in $O(m)$ time.*

**Proof.** Let $T = (\mathcal{A}(G), E_T)$ be an atom tree of $G$. We consider a rooted directed tree $T_r = (\mathcal{A}(G), U)$ obtained from $T$ by choosing an arbitrary root. Thus, $\Sigma_{XY \in E_T} |X \cap Y| = \Sigma_{(X,Y) \in U} |X \cap Y| \leq \Sigma_{(X,Y) \in U} |Y| \leq \Sigma_{Y \in \mathcal{A}(G)} |Y| \leq n + m$ by Property 6.
These sets can be computed by searching $T$ and computing $X \cap Y$ in $O(|Y|)$ time when reaching $Y$ from its neighbor $X$, and therefore in $O(m)$ time.by Property 6. □

*α-acyclic hypergraphs.* A *simple hypergraph*, or *hypergraph* for short, is a structure $H = (V, \mathcal{E})$, where $V$ is its vertex set and $\mathcal{E}$ is a set of non-empty subsets of $V$, called the *hyperedges* of $H$, whose union is equal to $V$. A hypergraph is a *clutter* if the elements of $\mathcal{E}$ are pairwise non-inclusive. Its *line graph*, denoted by $L(H)$, is the intersection graph of $\mathcal{E}$. Its *2-section graph*, denoted by $2SEC(H)$, is the graph whose vertex set is $V$ and whose edges are the pairs of $V$ that are contained in a hyperedge of $H$. $H$ is *connected* if $L(H)$ is connected, or equivalently, if $2SEC(H)$ is connected. We denote by $p$ the number of hyperedges of a hypergraph. Let $(v_1, \ldots, v_n)$ be an ordering of $V$, and let $(X_1, \ldots, X_p)$ be an ordering of $\mathcal{E}$. The *incidence matrix* of $H$ w.r.t. these orderings is the $n \times p$ matrix $M = (m_{i,j})$, such that, for each $i \in [1, n]$ and each $j \in [1, p]$, $m_{i,j} = 1$ if $v_i \in X_j$ and 0 otherwise.

A *join tree* of $H$ is a tree $T$ whose node set is $\mathcal{E}$, such that, for each vertex $x$ of $H$, the set $\mathcal{E}_x$ of nodes of $T$ containing $x$ induces a subtree of $T$, or equivalently, such that, for each pair $\{X, Y\}$ of $\mathcal{E}$, $X \cap Y$ is a subset of each node of $P_T(X, Y)$. $H$ is *α-acyclic* if it has a join tree.

A join tree of an $α$-acyclic hypergraph $H$ can be computed in $O(s)$ time, where $s$ is the sum of the sizes of the hyperedges of $H$ [27].

**Property 8.** *Let $H = (V, \mathcal{E})$ be an α-acyclic hypergraph, and let G be the 2-section graph $2SEC(H)$. Then, G is chordal, and if, moreover, H is a clutter, then $\mathcal{E} = \mathcal{K}(G)$ (i.e., the set $\mathcal{E}$ of hyperedges of H is equal to the set $\mathcal{K}(G)$ of maximal cliques of G).*

It follows that the number of hyperedges of a clutter is bounded by the number of its vertices, since a chordal graph has, at most, $n$ maximal cliques. The number of hyperedges of an $\alpha$-acyclic hypergraph that is not a clutter may be exponential in the number of vertices.

A join tree of an $\alpha$-acyclic hypergraph $H = (V, \mathcal{E})$ can be defined from its *weighted line graph*, where weights are defined as follows. The *set associated with* a pair $\{X, Y\}$ of $\mathcal{E}$ is $X \cap Y$, and its *weight*, denoted by $w(XY)$, is $|X \cap Y|$. Let $K$ be a graph whose node set is $\mathcal{E}$. The weight of $K$ is the sum of the weights of its edges. When considered a weighted graph, $K$ is denoted by $K_w$. Thus, $L_w(H)$ denotes the weighted line graph of $H$.

**Characterization 4** ([28]). *Let $H = (V, \mathcal{E})$ be an $\alpha$-acyclic (resp. connected $\alpha$-acyclic) hypergraph. Then, the join trees of $H$ are the maximum weight spanning trees of the weighted complete graph on $\mathcal{E}$ (resp. of $L_w(H)$).*

In particular, the atom trees of a connected graph $G$ are the maximum weight spanning trees of the weighted intersection graph of the atoms of $G$, which is proved in the case of chordal graph in [19] (and extends to any connected graph through the chordal graph $G^+$ by Property 4).

### 3. Atom Graphs

Atom graphs were used in [15] and were formally introduced in [16].

**Definition 3** ([16]). *The* atom graph *of a graph $G$, denoted by $AG(G)$, is the graph $(\mathcal{A}(G), E')$, where $\mathcal{A}(G)$ is the set of atoms and $E'$ the set of pairs $\{A, B\}$ of $\mathcal{A}(G)$, such that $A \cap B$ is a clique minimal $(A \setminus B)(B \setminus A)$-separator of $G$.*

**Example 2.** *Figure 2 shows the atom graph of the graph $G$ from Figure 1.*

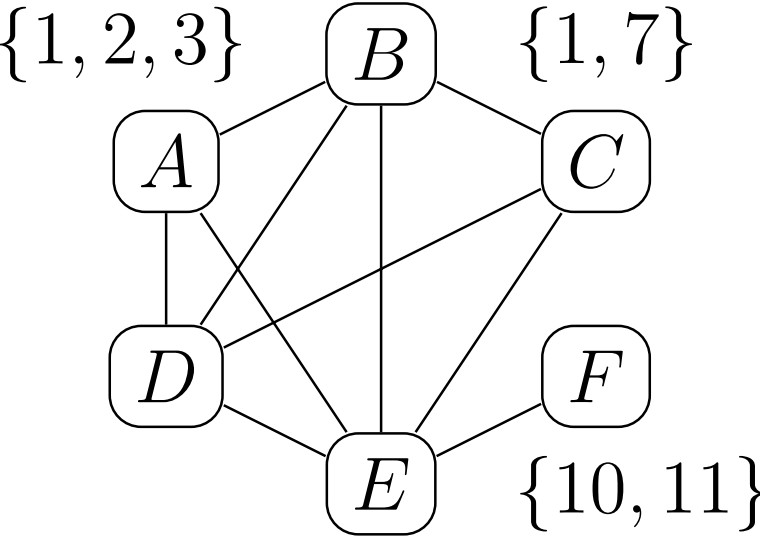

**Figure 2.** The atom graph of $G$. (the edge labels that are equal to 1 are omitted).

In the definition of the atom graph, the word 'clique' can be removed by Property 2, and the word 'minimal' can be removed by Property 5, which implies that, for each pairs $\{A, B\}$ of $\mathcal{A}(G)$, each one of $A \setminus B$ and $B \setminus A$ is a subset of a full component of $A \cap B$.

**Characterization 5.** *Let $A$ and $B$ be distinct atoms of a graph $G$. Then, $AB$ is an edge of $AG(G)$ if and only if $A \cap B$ is an $ab$-separator of $G$ for some $a \in A \setminus B$ and some $b \in B \setminus A$.*

The following property immediately follows from Properties 1 and 3.

**Property 9.** *The atom graph of a graph G is the atom graph of the chordal graph $G^+$ ($G^+$ is defined in Notation 1).*

Characterizations 6 and 7 below give relationships between the atom graph and the atom trees. They are both proved for chordal graphs in [21] and also apply to any connected graph through the chordal graph $G^+$ by Properties 1, 4, and 9.

**Characterization 6.** *The atom graph of a connected graph G is the union of all the atom trees of G.*

**Characterization 7.** *The atom trees of a connected graph G are the maximum weight spanning trees of the weighted atom graph of G.*

To compute the edges of the atom graph from an atom tree, we will use the following characterization from [22] for chordal graphs, which also applies to any connected graph through the chordal graph $G^+$ by Properties 1, 4, and 9.

**Characterization 8.** *Let G be a connected graph, let A and B be distinct atoms of G, and let T be an atom tree of G. Then, AB is an edge of $AG(G)$ if and only if there is an edge $A'B'$ on the path $P_T(A, B)$ between A and B in the tree T, such that $A \cap B = A' \cap B'$.*

## 4. Computing the Atom Graph

We know that, given a connected graph $G$, an atom tree of $G$ (and therefore the atoms of $G$) can be computed in linear time if $G$ is chordal and in $O(min(n^\omega \log n, nm, n(n + \overline{m})))$ time otherwise. To compute the set of edges of the atom graph of $G$, a naive algorithm consists of computing for each pair $\{A, B\}$ of atoms of $G$ the connected components of $G(V \setminus (A \cap B))$ and determining whether $A \setminus B$ and $B \setminus A$ are in different components, which can be performed in $O(m)$ time for each pair $\{A, B\}$ and therefore in $O(n^2m)$ time globally.

We will improve upon this to obtain a time that is no worse than that of computing an atom tree.

Our first algorithm starts with an atom tree and the inclusion relation between the separators represented by the edges, and adds all the extra edges required to construct the atom graph. Our second algorithm starts with the weighted intersection graph of the atoms and repeatedly determines the edges of weight $k$, which belong to the atom graph in decreasing order of $k$. Both algorithms run in $O(n^2)$ time given these inputs. When only the graph is given as input, we obtain a complexity of $O(min(n^\omega \log n, nm, n(n + \overline{m})))$ time, as will be detailed in this section.

We introduce the following parameters, which will be used in this section and in Section 6: $p$ denotes the number of atoms of $G$, $s$ the sum of their sizes, and for each atom tree $T$ of $G$, $s_\triangle(T)$ denotes the sum of the sizes of the symmetrical differences $X \triangle Y = (X \setminus Y) \cup (Y \setminus X)$ for each edge $XY$ of $T$.

**Notation 2.** *For each connected graph G, $p = |\mathcal{A}(G)|$, $s = \Sigma_{X \in \mathcal{A}(G)} |X|$, and for each atom tree, $T = (\mathcal{A}(G), E_T)$ of G $s_\triangle(T) = \Sigma_{XY \in E_T} |X \triangle Y|$.*

Note that $p \leq n$ since $G$ has at most $n$ atoms, and that $s \leq n + m$ since the sum of the sizes of the atoms of $G$ is bounded by $n + m$ by Property 6. The parameters $p$, $s$, and $s_\triangle(T)$ are introduced for two reasons. First, they will be used to extend the complexity results of this section to the context of $\alpha$-acyclic hypergraphs in Section 6 with appropriate extensions of the definitions of these parameters. Second, it can lead to a better complexity for graph classes for which these parameters have specific bounds.

As will be detailed in Section 4.1, we will also need the inclusion relation *sub* on the minimal separators represented by the edges of an atom tree.

**Definition 4.** *Let T be an atom tree of a connected graph. We call* subset relation *of T the relation sub in the set $E_T$ of edges of T defined by :* $\forall XY, X'Y' \in E_T \, sub(XY, X'Y') \Leftrightarrow X \cap Y \subseteq X' \cap Y'$.

We will show in Sections 4.1 and 4.2 the following complexity result:

**Theorem 1.** *The atom graph of a connected graph G can be computed :*
*(a) in $O(n^2)$ time from either an atom tree of G and its subset relation or the weighted intersection graph of the atoms of G,*
*(b) in $O(min(n^\omega, nm, n(n + \overline{m^+})))$ time from an atom tree of G,*
*(c) in $O(min(n^\omega, nm))$ time from the set of atoms of G,*
*(d) in $O(min(n^\omega \log n, nm, n(n + \overline{m})))$ time from G,*
*where $\overline{m^+}$ denotes the number of edges of $\overline{G^+}$ ($G^+$ is defined in Notation 1).*

For a chordal graph $H$, the atom graph can thus be computed in $O(min(n^\omega, nm, n(n + \overline{m})))$ time, since, in that case, $G^+ = G$, and an atom tree (clique tree) of $G$ can be computed in linear time.

Other approaches are possible but with no improvement of the time complexity. For instance, as the atom trees of a graph $G$ are obtained from the atom trees of a minimal triangulation $H$ of $G$ by merging the maximal cliques of $H$ into the atoms of $G$ [4], the atom graph of $G$ is obtained from the atom graph of $H$ by merging the same maximal cliques of $H$.

The different items of Theorem 1 are detailed in the following results: item (a) follows from Theorem 2 and Corollary 1, item (b) follows from item (c) and Theorem 3, item (c) follows from item (a) and Proposition 2, and item (d) follows from item (b) and from the fact that an atom tree of $G$ can be computed in $O(min(n^\omega \log n, nm, n(n + \overline{m})))$ time.

### 4.1. Algorithm Forest Join

Our first algorithm, Forest Join (Algorithm 1), is based on Characterization 9. Given an atom tree $T$, a minimal separator S is represented by one or several edges of $T$. If we remove these edges, we obtain a forest. Let us now further shrink this forest by removing the nodes that do not contain $S$. Any edge between two nodes of different trees of the resulting forest will correspond to an edge of the atom graph, which also represents S, and all the $S$ representatives are thus encountered.

To implement this remarkable property, our algorithm processes the edges of the atom tree one by one and computes the relevent nodes and edges with the help of relation *sub*.

**Characterization 9.** *Let G be a connected graph, let T be an atom tree of G, and let S be a minimal separator of G. Then, the edges of AG(G) associated with S are the pairs of nodes of T whose endpoints are in different connected components of $T(\mathcal{A}_S) - E_S$, where $\mathcal{A}_S$ is the set of nodes of T containing S, and $E_S$ is the set of edges of T associated with S.*

**Proof.** Let $\{X, Y\}$ be a pair of nodes of $T$. Let us show that $XY$ is an edge of $AG(G)$ associated with S if and only if $X$ and $Y$ are in different connected components of $T(\mathcal{A}_S) - E_S$, i.e., by Characterization 8, and the fact that $T(\mathcal{A}_S)$ is connected, that there is an edge $X'Y'$ of $P_T(X, Y)$, such that $S = X \cap Y = X' \cap Y'$ if and only if $P_T(X, Y)$ is a path in $T(\mathcal{A}_S)$ having an edge $X'Y'$ in $E_S$.
$\Rightarrow$: as $S = X \cap Y \, P_T(X, Y)$ is a path in $T(\mathcal{A}_S)$, and as $S = X' \cap Y'$, $X'Y'$ is in $E_S$.
$\Leftarrow$: as $X$ and $Y$ are in $\mathcal{A}_S$, $S \subseteq X \cap Y$. Hence, $S \subseteq X \cap Y \subseteq X' \cap Y' = S$, and therefore $S = X \cap Y = X' \cap Y'$. □

The algorithm Forest Join computes the edges of the atom graph of $G$ according to Characterization 9. Given an atom tree $T$ of $G$ and its subset relation *sub*, it scans the edges of $T$, and for each edge $AB$, it computes the set of edges of the atom graph associated with the minimal separator $S$ associated with $AB$ if it has not been computed yet, *i.e.,* if $AB$ does not belong to the set of edges computed so far.

It calls the algorithm Components (Algorithm 2), which computes the connected components of the forest $T(\mathcal{A}_S) - E_S$ defined in Characterization 9. The relation *sub* enables us to compute these components at no extra cost than a simple tree search: $T(\mathcal{A}_S)$ is the subtree of $T$ whose edges $XY$ are associated with supersets of $S$, i.e., satisfy $sub(AB, XY)$, and $E_S$ is the set of edges $XY$ of $T$ associated with $S$, i.e., such that $sub(AB, XY)$ and $sub(XY, AB)$.

In the algorithm Components, $k$ is the current number of connected components, $C_1, \ldots, C_k$ are the current components, *Queue* contains the nodes of $T$ that are reached but not processed yet, and for each reached node $X$, $numComp(X)$ is the index $i$ of the component $C_i$ containing $X$, and $pred(X)$ is the node of $T$ from which it has been reached (and which should not be processed again).

---

**Algorithm 1:** Forest Join.

> **input** : An atom tree $T = (\mathcal{A}, E_T)$ of a connected graph $G$ and its subset relation *sub*.
> **output:** The atom graph of $G$.
>
> $E' \leftarrow \varnothing$;
> **foreach** $AB \in E_T$ **do**
>   **if** $AB \notin E'$ **then**
>     // the edges associated with $A \cap B$ are not in $E'$ yet
>     $CompSet \leftarrow$ **Components**$(T, AB, sub)$;
>     **foreach** $\{C, C'\} \subseteq CompSet$ **do**
>       **foreach** $X \in C$ **do**
>         **foreach** $Y \in C'$ **do**
>           Add $XY$ to $E'$;
>         **end**
>       **end**
>     **end**
>   **end**
> **end**
> return $(\mathcal{A}, E')$;

---

**Example 3.** *Figure 3 shows an atom tree T of the graph G from Figure 1 and an execution of the algorithm Forest Join on T and its subset relation. It shows the forest $T(\mathcal{A}_S) - E_S$ for $S = \{1\}$, where the edges of the atom graph associated with S are represented by dotted lines. For each clique minimal separator S different from $\{1\}$, as $\mathcal{A}_S$ is of size 2, $AG(G)$ has a unique edge associated with S, which is also an edge of T. So, $AG(G)$ is obtained from T by adding the edges associated with $\{1\}$ that are not already present in T.*

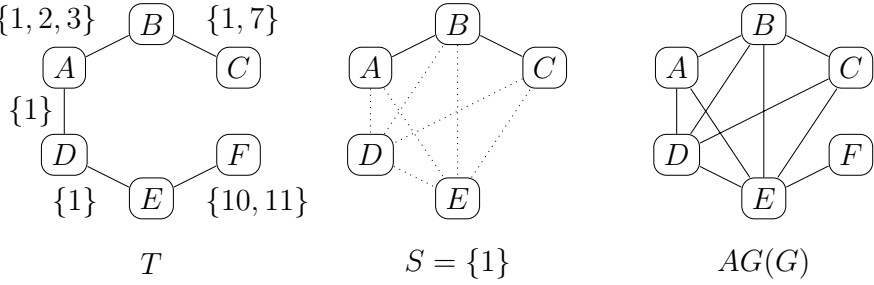

**Figure 3.** An execution of the algorithm Forest Join.

**Theorem 2.** *Given an atom tree of a connected graph G and its subset relation, the algorithm Forest Join computes the atom graph of G in $O(p^2)$ time, and therefore in $O(n^2)$ time.*

---

**Algorithm 2:** Components.

---

**input** : An atom tree $T$ of an connected graph, an edge $AB$ of $T$, and the subset
  relation $sub$ of $T$.

**output**: The set of connected component of $T(\mathcal{A}_S) - E_S$, where $S = A \cap B$, $\mathcal{A}_S$ is
  the set of nodes of $T$ containing $S$, and $E_S$ is the set of edges of $T$
  associated with $S$.

$k \leftarrow 1; C_1 \leftarrow \{A\}; numComp(A) \leftarrow 1; Queue \leftarrow \{A\};$
**while** $Queue \neq \varnothing$ **do**
  Remove a node $X$ from $Queue$;
  **foreach** $Y \in N_T(X)$ **do**
    **if** $(Y \neq pred(X)) \wedge sub(AB, XY)$ **then**
      **if** $sub(XY, AB)$ **then**
        // $XY$ associated with $S$, begin a new component
        $k \leftarrow k + 1; C_k \leftarrow \varnothing; i \leftarrow k;$
      **end**
      **else**
        $i \leftarrow numComp(X);$
      **end**
      Add $Y$ to $C_i$; $numComp(Y) \leftarrow i;$
      $pred(Y) \leftarrow X$; Add $Y$ to $Queue;$
    **end**
  **end**
**end**
return $\{C_1, \ldots, C_k\};$

---

**Proof.** The correctness follows from Characterization 9. Let us prove the time complexity.
As the algorithm Components runs in $O(p)$ time and is called less than $p$ times, it globally
costs $O(p^2)$ time. As an edge $XY$ is added to $E'$ at most once (when processing the first
edge of $T$ associated with $X \cap Y$), the number of edge additions to $E'$ is bounded by $p^2$.
Hence, the algorithm Forest Join runs in $O(p^2)$ time, and therefore in $O(n^2)$ time since $G$
has at most $n$ atoms ($p \leq n$). $\square$

To evaluate the time complexity of computing the atom graph of $G$ from an atom tree
$T$ of $G$ using the algorithm Forest Join, we need the time complexity of computing the
subset relation of $T$.

**Proposition 1.** *Given an atom tree of a connected graph, its subset relation can be computed in*
$O(min(n^\omega, ps))$ *time, and therefore in* $O(min(n^\omega, nm))$ *time.*

**Proof.** It follows from the proof of Property 7 that the sets $X \cap Y$ for each edge of $T$ can be
computed in $O(s)$ time. and that the sum of their sizes is bounded by $s$. As the inclusion of
$X \cap Y$ in $X' \cap Y'$ or not can be determined in $O(|X \cap Y|)$ time, the subset relation can be
computed in $O(s + ps)$ time, i.e., in $O(ps)$ time.
Alternatively, as $sub(XY, X'Y')$ is equivalent to $|X \cap Y| = |(X \cap Y) \cap (X' \cap Y')|$, it can be
evaluated in $O(1)$ time, and therefore in $O(p^2)$ time globally, provided that the values of
$|X \cap Y|$ and $|(X \cap Y) \cap (X' \cap Y')|$ have been pre-computed. The values of $X \cap Y$ and $|X \cap Y|$
for each edge $XY$ of $T$ can be computed in $O(s)$ time, and the values of $|(X \cap Y) \cap (X' \cap Y')|$
in $O((n + p)^\omega)$ time, since they are the elements of the product of the transposition of $M$ by
$M$, where $M$ is the $n \times (p - 1)$ incidence matrix of the (possibly non-simple) hypergraph
whose vertex set is $V$ and whose hyperedges are the sets $X \cap Y$ for each edge $XY$ of $T$.
Hence, this alternative complexity is in $O(p^2 + s + (n + p)^\omega)$ time, i.e., in $O((n + p)^\omega)$
time, since $p^2 \leq (n + p)^2$, $s \leq np \leq (n + p)^2$ and $2 \leq \omega$.
We obtain a complexity in $O(min((n + p)^\omega, ps))$ time, i.e., in $O(min(n^\omega, ps))$ time, since
$p \leq n$ and therefore in $O(min(n^\omega, nm))$ time, since $s \leq n + m$ by Property 6. $\square$

It follows that the atom graph can be computed from an atom tree in $O(min(n^\omega, nm))$ time.

We will now discuss using the 2-pairs of the graph $G^+$ defined in Notation 1 to obtain an alternative complexity in $O(n(n + \overline{m^+}))$ time, where $\overline{m^+}$ is the number of edges of the complement of $G^+$, through the following lemma.

**Lemma 1.** *Let T be atom tree of a connected graph. Then, $s_\triangle(T) \leq n + t$, where t is the number of 2-pairs of $G^+$.*

**Proof.** We consider a rooted directed tree $T_r = (\mathcal{A}(G), U)$ obtained from $T$ by choosing an arbitrary root. Thus, $\Sigma_{XY \in E_T}|X \triangle Y| = \Sigma_{(X,Y) \in U}|X \triangle Y|$. $\Sigma_{(X,Y) \in U}|Y \setminus X| \leq n$, since each vertex $x$ of $G$ belongs to $Y \setminus X$ for at most one edge of $T_r$, namely the edge $(X, Y)$, such that $Y$ is the root of the subtree of $T_r$ induced by the nodes containing $x$. Hence, it is sufficient to show that $\Sigma_{(X,Y) \in U}|X \setminus Y| \leq t$. It is shown in [4] that, if $G$ is chordal, then this sum is bounded by the number of 2-pairs of $G$. So, it is bounded by $t$ since $G^+$ is chordal and by Property 4 $T$ is also an atom tree of $G^+$.　□

**Theorem 3.** *The atom graph of a connected graph G can be computed from an atom tree T of G in $O(p(n + s_\triangle(T)))$ time, and therefore in $O(n(n + \overline{m^+}))$ time, where $\overline{m^+}$ is the number of edges of the complement of $G^+$.*

**Proof.** We consider the variant of the algorithm Forest Join, where the subset relation *sub* is not given as an input, and "*sub(AB, XY)*" and "*sub(XY, AB)*" in the algorithm Components are replaced as follows. Condition $sub(XY, AB)$ can be replaced by $|A \cap B| = |X \cap Y|$, since, in the algorithm, $XY$ satisfies $sub(AB, XY)$. The values of $|X \cap Y|$ for each edge $XY$ of $T$ can be pre-computed in $O(np)$ time. Let $S = A \cap B$. As $S$ is a subset of $X$ in the algorithm, and condition $sub(AB, XY)$ is equivalent to $(X \setminus Y) \cap S = \emptyset$, which can be evaluated in $O(|X \setminus Y|)$ time, provided that the sets $X \cap Y$, $X \setminus Y$ and $Y \setminus X$ for each edge $XY$ of $T$ have been pre-computed, which can be done in $O(np)$ time. Thus, we add to the time complexity of Algorithm Forest Join in $O(p^2)$ a pre-computation time in $O(np)$ and $O(s_\triangle(T))$ time per call to components. We obtain a time complexity in $O(p^2 + np + p * s_\triangle(T))$, and therefore in $O(p(n + s_\triangle(T)))$, since $p - 1 \leq s_\triangle(T)$ (because the nodes of $T$ are pairwise distinct). We conclude with Lemma 1　□

The 2-pairs of a chordal graph are closely related to its atom graph.

**Characterization 10.** *Let G be a connected chordal graph, and let $\{x, y\}$ be a pair of vertices of G. Then, xy is a 2-pair of G if and only if there is an edge KL of AG(G), such that $x \in K \setminus L$ and $y \in L \setminus K$.*

**Proof.** $\Rightarrow$: let $S = N(x) \cap N(y)$. As $S$ is a minimal separator of $G$ and $G$ is chordal, $S$ is a clique. Let $K$ (resp. $L$) be a maximal clique containing $\{x\} \cup S$ (resp. $\{y\} \cup S$). $S \subseteq K \cap L \subseteq (N(x) \cup \{x\}) \cap (N(y) \cup \{y\}) = N(x) \cap N(y) = S$. Hence $S = K \cap L$, and therefore, $KL$ is an edge of $AG(G)$ with $x \in K \setminus L$ and $y \in L \setminus K$.
$\Leftarrow$: let $S = K \cap L$. As $KL$ is an edge of $AG(G)$, $S$ is a minimal $xy$-separator. As $G$ is chordal $K$ and $L$ are cliques, so $S \subseteq N(x) \cap N(y)$, and as $S$ is an $xy$-separator $N(x) \cap N(y) \subseteq S$. Hence, $S = N(x) \cap N(y)$, and therefore $xy$ is a 2-pair of $G$.　□

It follows that the number of 2-pairs of a connected chordal graph $G$ is bounded by the sum of the products $|K \setminus L| * |L \setminus K|$ for each edge $KL$ of its atom graph. In particular, in a graph class (of non-necessarily chordal graphs) in which the values of $|A \setminus B|$ (and $|B \setminus A|$) for each edge $AB$ of the atom graph are bounded by a given constant, for instance, if the sizes of the atoms are bounded by a constant, the atom graph can be computed from an atom tree in $O(n(n + m'))$ time, where $m'$ is the number of edges of the computed atom graph. The number of 2-pairs is not equal, in general, to the sum of the products

$|K \setminus L| * |L \setminus K|$ for each edge $KL$ of its atom graph, since a 2-pair may be associated with several edges of the atom graph. Considering the same relation between the 2-pairs and the edges of an atom tree $T$ of $G$, a pair $\{x, y\}$ associated with an edge $KL$ of $T$ is a 2-pair, since $KL$ is an edge of the atom graph, but the converse does not hold. Contrary to the atom graph, $\{x, y\}$ can be associated with at most one edge of $T$, namely the unique edge connecting the subtrees of $T$ induced by the sets of nodes containing $x$ and $y$, respectively, which are necessarily disjoint and at distance 1 from each other in $T$.

Thus, the atom graph of $G$ can be computed from an atom tree of $G$ in $O(min(n^\omega, nm, n(n + \overline{m})))$. This time complexity considers the worst case, where the algorithm Components searches the whole tree $T$, whereas it only searches the set $\mathcal{A}_S$ and its neighborhood, which may be very small w.r.t. the set of nodes of $T$. It may be more efficient in practice to execute the algorithm Forest Join without pre-computing the subset relation *sub* and directly evaluate $sub(AB, XY)$ and $sub(XY, AB)$ when needed.

### 4.2. Algorithm AG-Max-Weight

Our second algorithm, AG-max-weight (Algorithm 3), takes as input the weighted intersection graph of the atoms (which, in the case of a chordal graph, is the clique graph) and repeatedly adds the edges of weight $k$ in decreasing order of $k$.

By Characterization 4, the atom trees of a connected graph $G$ are the maximum weight spanning trees of the weighted intersection graph of the atoms of $G$. We will present a general algorithm computing the union of the maximum weight spanning trees of $G_w$ for each weighted connected graph $G_w$ with natural integer weights on the edges. This general algorithm, called Union-max-weight (Algorithm 4), is inspired by the following algorithm from Kruskal, which computes a *minimum* weight spanning tree of $G_w$ : initialize graph $T'$ as edgeless and for each edge $xy$ of $G_w$ in *increasing* order of weight, add $xy$ to $T'$ if and only if $x$ and $y$ are in different connected components of $T'$. As we want to compute a *maximum* weight spanning tree, we will process the edges in *decreasing* order of weight; the algorithm computes each maximum weight spanning tree of $G_w$. Thus, an edge $xy$ of weight $k$ may be added to $T'$ by this last algorithm if and only if $x$ and $y$ are in different connected components of $T'$ just after processing the edges of weight at least $k + 1$. These components are independent of the graph $T'$ computed so far by item a) of Lemma 2 below.

**Lemma 2.** *Let $G_w = (V, E, w)$ be a weighted connected graph with natural integer weights on the edges, let $T$ be a maximum weight spanning tree of $G$, let $UM$ be the union of the maximum weight spanning trees of $G$ and for a natural integer $k$, and let $G_k$ (resp. $T_k$, $UM_k$) be the graph whose vertex set is $V$ and whose edges are the edges of $G$ (resp. $T$, $UM$) of weight at least $k$. Then,*
*(a) $G_k$, $T_k$ and $UM_k$ have the same connected components;*
*(b) the edges of $UM$ of weight $k$ are the edges of $G$ of weight $k$ whose endpoints are in different connected components of $UM_{k+1}$.*

**Proof.** (a) As each connected component of $T_k$ is a subset of a connected component of $UM_k$, which is itself a subset of a connected component of $G_k$, it is sufficient to show that each connected component of $G_k$ is a subset of a connected component of $T_k$, or equivalently, that, for each edge $xy$ of $G_k$, $P_T(x, y)$ is a path in $T_k$. Let $xy$ be an edge of $G_k$. For each edge $x'y'$ of $P_T(x, y)$, $w(xy) \leq w(x'y')$ (otherwise, $(T - \{x'y'\}) + \{xy\}$ would be a spanning tree of $G$ of strictly greater weight than $T$), so $P_T(x, y)$ is a path in $T_k$.
(b) Let $xy$ be an edge of $G$ of weight $k$. Let us show that $xy$ is an edge of $UM$ if and only if $x$ and $y$ are in different connected components of $UM_{k+1}$. We assume that $xy$ is an edge of $UM$. Let $T$ be a maximum weight spanning tree of $G$, such that $xy$ is an edge of $T$. $x$ and $y$ are in different connected components of $T_{k+1}$, and therefore of $UM_{k+1}$ by a). Conversely, we assume that $x$ and $y$ are in different connected components of $UM_{k+1}$. Let $T$ be a maximum weight spanning tree of $G$. As $x$ and $y$ are in different connected components of $T_{k+1}$, there is an edge $x'y'$ of $P_T(x, y)$ of weight at most $k$. Then, $(T - \{x'y'\}) + \{xy\}$ is a maximum weight spanning tree of $G$, and therefore $xy$ is an edge of $UM$. $\square$

Item (b) of Lemma 2 provides an inductive definition of the edges of weight $k$ of the union of the maximum weight spanning trees, and therefore a simple iterative algorithm to compute them. Thus, algorithm Union-max-weight computes the union of the maximum weight spanning trees of $G$ by initializing a set $F$ with the empty set and adding to $F$, for each weight value $k$ in decreasing order, the edges $xy$ of $G$ of weight $k$ such that $x$ and $y$ are in different connected components of the graph $(V, F)$ in its state just after adding the edges of weight strictly greater than $k$.

In the algorithm Union-max-weight, $k$ is the current value of weight, the sets $C_i$ are the connected components of the graph $(V, F)$ in its state at the beginning of iteration $k$ and for each vertex $x$, and $numComp(x)$ is the index $i$ of the component $C_i$ containing $x$. The algorithm is similar to the "maximum weight" variant of Kruskal's algorithm, the difference being that Kruskal's algorithm considers the connected components of the graph (tree) $(V, F)$ being computed in its current state instead of in its state at the beginning of iteration $k$, and therefore would update the variables $C_i$ and $numCom$ just after each addition of an edge to $F$. It follows that the algorithms and complexity results already published on Kruskal's algorithm hold for the computation of the union of the maximum weight spanning trees. In particular, the complexity can be improved by using a sophisticated UNION-FIND data structure. However, the simple algorithm presented here is sufficient to compute the atom graph in $O(n^2)$ time.

---

**Algorithm 3:** AG-max-weight.

> **input** :The weighted intersection graph $IG_w$ of the atoms of a connected graph $G$.
> **output**:The atom graph of $G$.
>
> return **Union-max-weight**$(IG_w)$;

---

**Algorithm 4:** Union-max-weight.

> **input** :A weighted-connected graph $G_w = (V, E, w)$, with natural integer
> weights on the edges
> **output**:The union of the maximum weight spanning trees of $G_w$.
>
> Compute the maximum weight $w_{max}$ of an edge of $G_w$ and for each $k$ in $[0, w_{max}]$
>   the set $E_k$ of edges of $G_w$ of weight $k$;
> $i \leftarrow 0$;
> **foreach** $x \in V$ **do**
>   | $i \leftarrow i + 1$; $C_i \leftarrow \{x\}$; $numComp(x) \leftarrow i$;
> **end**
> $F \leftarrow \varnothing$;
> **foreach** $k = w_{max}$ *downto* $0$ **do**
>   **foreach** $xy \in E_k$ **do**
>     **if** $numComp(x) \neq numComp(y)$ **then**
>       | Add $xy$ to $F$;
>     **end**
>   **end**
>   **foreach** $xy \in E_k$ **do**
>     **if** $numComp(x) \neq numComp(y)$ **then**
>       $i \leftarrow numComp(x)$; $j \leftarrow numComp(y)$; $C_i \leftarrow C_i \cup C_j$;
>       **foreach** $z \in C_j$ **do**
>         | $numComp(z) \leftarrow i$;
>       **end**
>     **end**
>   **end**
> **end**
> return $(V, F)$;

**Example 4.** *Figure 4 shows the weighted intersection graph of the atoms of the graph G from Figure 1 and an execution of the algorithm AG-max-weight, i.e., the algorithm Union-max-weight, on this weighted graph. It shows the state of the computed graph before and after adding the edges of weight 1.*

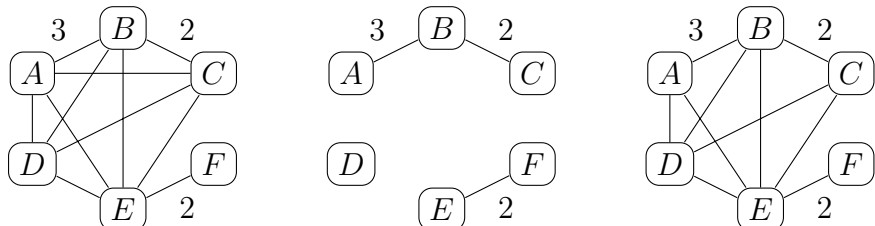

**Figure 4.** An execution of the algorithm AG-max-weight (the edge labels that are equal to 1 are omitted).

**Theorem 4.** *Given a weighted connected graph $G_w = (V, E, w)$ with natural integer weights on the edges, the algorithm Union-max-weight computes the union of the maximum weight spanning trees of $G_w$ in $O(w_{max} + n^2)$ time, where $w_{max}$ is the maximum weight of an edge of $G_w$.*

**Proof.** It follows from Lemma 2 that the property $P$ defined below is an invariant of the main foreach loop, using the notation $UM_k$ of this lemma,
$P : UM_k = (V, F)$ and $\forall x \in V$ ($C_{numComp(x)}$ is the connected component of $UM_k$ containing $x \wedge \forall y \in C_{numComp(x)}$ $numComp(x) = numComp(y)$),
which proves the correctness of the algorithm.
Let us prove its time complexity. $w_{max}$ and the sets $E_k$ can be computed and scanned in the internal for each loop in $O(w_{max} + m)$ time by storing the elements of $E_k$ at index $k$ of an array. The first internal for each loop runs in $O(m)$ time globally, and the second one in $O(n^2)$ time globally, since merging two connected components is in $O(n)$ time and is performed $n - 1$ times. Hence, the algorithm runs in $O(w_{max} + n^2)$ time. $\square$

**Corollary 1.** *Given the weighted intersection graph of the atoms of a connected graph G, the algorithm AG-max-weight computes the atom graph of G in $O(n + p^2)$ time and therefore in $O(n^2)$ time.*

To evaluate the time complexity of computing the atom graph of *G* from the set of atoms of *G* using the algorithm AG-max-weight, we need the time complexity of computing the the weighted intersection graph of the atoms of *G*.

**Proposition 2.** *Given the set of atoms of a connected graph G, the weighted intersection graph of the atoms of G can be computed in $O(min(n^\omega, ps))$ time and therefore in $O(min(n^\omega, nm))$ time.*

**Proof.** As $|X \cap Y|$ can be computed in $O(|Y|)$ time, computing $|X \cap Y|$ for each pair $\{X, Y\}$ of atoms of *G* is in $O(ps)$ time. Alternatively, these values can be computed in $O((n + p)^\omega)$ time, since they are the elements of the product of the transposition of *M* by *M*, where *M* is the $n \times p$ incidence matrix of the hypergraph $(V, \mathcal{A}(G))$ (which will be called the atom hypergraph of *G* in Section 5), i.e., in $O(n^\omega)$ time since $p \leq n$. We obtain a time complexity in $O(min(n^\omega, ps))$ and therefore in $O(min(n^\omega, nm))$, since $s \leq n + m$ by Property 6. $\square$

It follows that the atom graph can be computed from the set of atoms in $O(min(n^\omega, nm))$ time.

## 5. Atom Hypergraph

In this section, we define the atom hypergraph of a graph and relate it to the more general notion of *α*-acyclic hypergraph.

**Definition 5.** *Let $G = (V, E)$ be a graph. The* atom hypergraph *of $G$ is the hypergraph $H_A(G) = (V, \mathcal{A}(G))$.*

Thus, the atom trees of a connected graph are the join trees of its atom hypergraph. We recall that, for each hypergraph $H$, $2SEC(H)$ is the graph whose vertex set is the vertex set of $H$ and whose edges are the pairs of vertices that are contained in a hyperedge of $H$

**Characterization 11.** *An hypergraph is the atom hypergraph of a connected graph if and only if it is a connected $\alpha$-acyclic clutter, and in that case, it is the atom hypergraph of the graph $2SEC(H)$, which is a connected chordal graph.*

**Proof.** The atom hypergraph of a connected graph $G$ is connected (since $G$ is and each edge of $G$ is contained in an atom of $G$), $\alpha$-acyclic (since $G$ has an atom tree) and a clutter (by definition of atoms). Conversely, if $H$ is a connected $\alpha$-acyclic clutter, then by Property 8, it is the atom hypergraph of the graph $2SEC(H)$, which is chordal, and which is connected, since $H$ is.　□

Note that, if $H$ is the atom hypergraph of $G$, then $2SEC(H)$ is the graph $G^+$ defined in Notation 1. Thus, we refind that $G^+$ is chordal and has the same atoms as $G$ (Property 3).

**Definition 6.** *The* union join graph *of an $\alpha$-acyclic hypergraph $H$, denoted by $UJ(H)$, is the union of its join trees.*

As the atom graph of a connected graph $G$ is the union of its atom trees by Characterizations 6, we have the following property.

**Property 10.** *The atom graph of a connected graph is the union join graph of its atom hypergraph.*

As a generalization of Characterizations 8, the union join graph of an $\alpha$-acyclic hypergraph $H$ can be computed from a join tree of $H$ by the following operation *tuj*, where *tuj* stands for "to union join".

**Definition 7.** *For each join tree $T = (\mathcal{E}, E_T)$ of a hypergraph, $tuj(T)$ is the graph whose node set is $\mathcal{E}$ and whose edges are the pairs $\{X, Y\}$ of $\mathcal{E}$, such that there is an edge $X'Y'$ of $P_T(X, Y)$, such that $X \cap Y = X' \cap Y'$ (or equivalently $X' \cap Y' \subseteq X \cap Y$).*

**Characterization 12.** *For each $\alpha$-acyclic hypergraph $H$ and each join tree $T$ of $H$, $UJ(H) = tuj(T)$.*

**Proof.** Let $H = (V, \mathcal{E})$ and let $\{X, Y\} \subseteq \mathcal{E}$. Let us show that $XY$ is an edge of $UJ(H)$ if and only if $XY$ is an edge of $tuj(T)$.
$\Rightarrow$ : let $T'$ be a join tree of $H$, such that $XY$ is an edge of $T'$, and let $\mathcal{E}_X$ (resp. $\mathcal{E}_Y$) be the connected component of $T' - \{XY\}$ containing $X$ (resp. $Y$). As $X \in \mathcal{E}_X$ and $Y \in \mathcal{E}_Y$, there is an edge $X'Y'$ of $P_T(X, Y)$ such that $X' \in \mathcal{E}_X$ and $Y' \in \mathcal{E}_Y$. As $T'$ is a join tree and $XY$ is an edge of $P_{T'}(X', Y')$, $X' \cap Y' \subseteq X \cap Y$. Hence, $XY$ is an edge of $tuj(T)$.
$\Leftarrow$ : let $X'Y'$ be an edge of $P_T(X, Y)$, such that $X \cap Y = X' \cap Y'$, and let $T'$ be the graph $(T - \{X'Y'\}) + \{XY\}$. $T'$ is a tree with the same weight as $T$ (since $w(XY) = w(X'Y')$), so by Characterization 4, $T'$ is also a join tree of $H$, and therefore, $XY$ is an edge of $UJ(H)$.　□

Thus, we refind Characterization 8 from Property 10 and Characterization 12. Conversely, Characterization 12 can be deduced from Characterization 8 and Property 11 below, which shows that any $\alpha$-acyclic hypergraph is an atom hypergraph up to isomorphism.

**Notation 3.** *Let $\mathcal{E}$ and $\mathcal{E}'$ be two sets and let $f$ be a one-to-one mapping from $\mathcal{E}$ to $\mathcal{E}'$. For each graph $K = (\mathcal{E}, E_K)$, $f(K)$ denotes the graph obtained from $K$ by isomorphism $f$, i.e., $f(K) = (\mathcal{E}', \{f(X)f(Y), XY \in E_K\})$.*

**Property 11.** *Let $H = (V, \mathcal{E})$ be an $\alpha$-acyclic hypergraph. Then there is a connected chordal graph $G = (V', E_G)$ and a one-to-one mapping $f$ from $\mathcal{E}$ to $\mathcal{A}(G)$ such that :*
*(1) for each tree $T = (\mathcal{E}, E_T)$, $T$ is a join tree of $H$ if and only if $f(T)$ is an atom tree of $G$;*
*(2) $AG(G) = f(UJ(H))$;*
*(3) If $H$ is connected, then for each pair $\{X, Y\}$ of $\mathcal{E}$, $f(X) \cap f(Y) = X \cap Y$; otherwise, there is an element a of $V'$, such that, for each pair $\{X, Y\}$ of $\mathcal{E}$, $f(X) \cap f(Y) = (X \cap Y) + \{a\}$;*
*(4) for each join tree $T$ of $H$, $tuj(f(T)) = f(tuj(T))$.*

**Proof.** By Characterization 11, it is sufficient to find a connected $\alpha$-acyclic clutter $H' = (V', \mathcal{E}')$ and a one-to-one mapping $f$ from $\mathcal{E}$ to $\mathcal{E}'$, such that: 1) for each tree $T = (\mathcal{E}, E_T)$, $T$ is a join tree of $H$ if and only if $f(T)$ is a join tree of $H'$; 2) $UJ(H') = f(UJ(H))$, and items 3) and 4). Let $\mathcal{E}'$ be defined from $\mathcal{E}$ by adding a new specific element $a_X$ to each element of $\mathcal{E}$, which is not inclusion-maximal in $\mathcal{E}$, and adding a new common element $a$ to each element of $\mathcal{E}$ if $H$ is not connected. Let $f$ map each element of $\mathcal{E}$ to the element of $\mathcal{E}'$ obtained from it, let $V' = \cup_{X \in \mathcal{E}'} X$, and let $H' = (V', \mathcal{E}')$. By definition, $H'$ is a connected clutter satisfying 3). As for each added element $a_X$ (resp. $a$), the set of elements of $\mathcal{E}'$ containing it is reduced to $\{X\}$ (resp. equal to $\mathcal{E}'$), $H'$ is $\alpha$-acyclic and satisfies 1); 2) follows from 1), and 4) follows from 3). □

Thus, we can deduce from properties of $\alpha$-acyclic hypergraphs (proved from the definition of $\alpha$-acyclicity) properties of atom graphs, and conversely, we can deduce from properties of atom graphs (proved from properties of the minimal separators of the underlying graph) properties of general $\alpha$-acyclic hypergraphs. This double approach helps to increase knowledge in both domains of atom graphs and $\alpha$-acyclic hypergraphs, as some properties are easier to see in one of these domains than in the other one.

Let us consider the case of a disconnected $\alpha$-acyclic hypergraph $H$. A join tree of $H$ is obtained from the disjoint union of join trees of its connected components by adding "empty edges", i.e., edges associated with the empty set, to make it into a tree, and the union join graph of $H$ is obtained from the disjoint union of the union join graphs of its connected components by adding all edges (which are empty edges) between these union join graphs. Alternatively, if we omit the empty edges, a join tree becomes a forest called *join forest* in [27], whose connected components are the join trees of its connected components, and the union join graph becomes the union of its join forests, whose connected components are the union join graphs of its connected components.
These two alternatives also exist in the graph and minimal separator approach, which correspond to two different definitions of separators in a disconnected graph. According to the definition of separators given in this paper, as the empty set is not a separator and the edges of an atom tree represent the minimal separators, an atom tree naturally extends to an atom forest of a (not necessarily connected) graph, which is the join forest of its atom hypergraph, and its atom graph (Definition 3) is the union of these join forests. According to an alternative definition of separators, which is given, for instance, in [2], a set $S$ is an $ab$-separator of $G$ if $a$ and $b$ are in different connected components of $G(V \setminus S)$, whether $G$ is connected or not. It follows that the empty set is the unique minimal $ab$-separator of $G$ if $a$ and $b$ are in different connected components of $G$. Thus, according to this alternative definition, an atom tree of a (not necessarily connected) graph is a join tree of its atom hypergraph and its atom graph (Definition 3 again) is the union join graph of its atom hypergraph. In both alternatives, the results for a connected graph naturally extend to a (non necessarily connected) graph using the appropriate definition of separators, as well as to (not necessarily connected) $\alpha$-acyclic hypergraphs, as will be seen in Section 6.

## 6. Computing the Union Join Graph

Algorithms and complexity results of Section 4 extend to the computation of the union join graph of an $\alpha$-acyclic hypergraph. They immediately extend to a connected $\alpha$-acyclic clutter $H$, since, in that case, $H$ is the atom hypergraph of $2SEC(H)$ by Characterization 11. The algorithms still hold for any $\alpha$-acyclic hypergraph, since the proofs of their correctness

do. It is also the case for the complexity bounds in function of parameters $n$, $p$, $s$, and $s_\triangle(T)$, whose definitions naturally extend to $\alpha$-acyclic hypergraphs as follows.

**Notation 4.** *For each $\alpha$-acyclic hypergraph $H = (V, \mathcal{E})$, $n = |V|$, $m$ is the number of edges of $2SEC(H)$, $\overline{m}$ is the number of edges of its complement, $p = |\mathcal{E}|$, $s = \Sigma_{X \in \mathcal{E}} |X|$, and for each join tree $T = (\mathcal{E}, E_T)$ of $H$ $s_\triangle(T) = \Sigma_{XY \in E_T} |X \triangle Y|$.*

We recall that $\omega$ is a real number, such that $O(n^\omega)$ is the best known time complexity of matrix multiplication. The subset relation of a join tree is defined as that of an atom tree (see Definition 4).

**Theorem 5.** *The union join graph of an $\alpha$-acyclic hypergraph H can be computed*
*(a) in $O(p^2)$ time from a join tree of H and either its subset relation or the weighted line graph of H;*
*(b) in $O(n + p^2)$ time from the weighted line graph of H;*
*(c) in $O(min((n + p)^\omega, ps, p(n + s_\triangle(T))))$ time from H, where T is a join tree of H (which is computed along with the union join graph of H).*

**Proof.** Item a) follows from Theorem 6 and Theorem 7, item b) follows from Corollary 2, item c) follows from item b) and Proposition 4 for the $O(min((n + p)^\omega, ps))$ bound, as well as from the extension of Theorem 3 to $\alpha$-acyclic hypergraphs and the fact that a join tree can be computed in $O(s)$ time [27] for the $O(p(n + s_\triangle(T)))$ bound. $\square$

The four results below extend Theorem 2, Proposition 1, Corollary 1 and Proposition 2, respectively. The complexity bound $n^\omega$ is replaced by $(n + p)^\omega$, which is the original bound appearing in the proofs of the concerned results and has been simplified into $n^\omega$ since $p \leq n$ in the case of the atom graph (a graph has at most $n$ atoms).

**Theorem 6.** *Given a join tree of an $\alpha$-acyclic hypergraph H and its subset relation, the algorithm Forest Join computes the union join graph of H in $O(p^2)$ time.*

**Proposition 3.** *Given a join tree of an $\alpha$-acyclic hypergraph, its subset relation can be computed in $O(min((n + p)^\omega, ps))$ time.*

**Corollary 2.** *(of Theorem 4) Given the weighted line graph of an $\alpha$-acyclic hypergraph H, the algorithm Union-max-weight computes the union join graph of H in $O(n + p^2)$ time.*

**Proposition 4.** *Given an $\alpha$-acyclic hypergraph, its weighted line graph can be computed in $O(min((n + p)^\omega, ps))$ time.*

We present now the algorithm UJ-min-weight (Algorithm 5), which is an alternative to the algorithm Union-max-weight, computing the union join graph in $O(p^2)$ time instead of $O(n + p^2)$ time but requiring a join tree of $H$ as input in addition to the weighted line graph of $H$. This algorithm can obviously be used to compute the atom graph of a connected graph $G$ from an atom tree and the weighted intersection graph of the atoms of $G$ in $O(p^2)$ time and therefore in $O(n^2)$ time, which is already done by the algorithm AG-max-weight with less input. The algorithm follows from Characterization 13 below, which is an immediate consequence of the characterization of $UJ(H)$ as $tuj(T)$ (Characterization 12). The algorithm computes for each pair $\{X, Y\}$ of hyperedges of $H$ the minimum weight of an edge of the path in $T$ between $X$ and $Y$ and stores it in the variables $minWeight(X, Y)$ and $minWeight(Y, X)$ to be used later in the execution.

**Characterization 13.** *Let $H = (V, \mathcal{E})$ be an $\alpha$-acyclic hypergraph, let T be a join tree of H, and let $\{X, Y\} \subseteq \mathcal{E}$. Then, XY is an edge of $UJ(H)$ if and only if its weight is the minimum weight of an edge of $P_T(X, Y)$.*

**Proof.** Let $w_{min}$ be the minimum weight of an edge of $P_T(X, Y)$. As $X \cap Y$ is a subset of $X' \cap Y'$ for each edge $X'Y'$ of $P_T(X, Y)$, since $T$ is a join tree, $w(XY) \le w_{min}$, and $w(XY) = w_{min}$ if and only there is an edge $X'Y'$ of $P_T(X, Y)$, such that $X \cap Y = X' \cap Y'$, i.e., if and only if $XY$ is an edge of $UJ(H)$ by Characterization 12. □

---

**Algorithm 5:** UJ-min-weight.

---

**input** : A join tree $T = (\mathcal{E}, E_T)$ and the weighted line graph $L_w(H)$ of an $\alpha$-acyclic hypergraph $H$.

**output**: The union join graph of $H$.

// in the following, $w(e) = 0$ if $e$ is a non-edge of $L_w(H)$;
Choose a node $X$ of $T$;
$Reached \leftarrow \{X\}; Queue \leftarrow \{X\}; E' \leftarrow E_T$;
**while** $Queue \ne \emptyset$ **do**
　　Remove a node $X$ from $Queue$;
　　**foreach** $Y \in N_T(X)$ **do**
　　　　**if** $Y \notin Reached$ **then**
　　　　　　$minWeight(X, Y) \leftarrow w(XY); minWeight(Y, X) \leftarrow w(XY)$;
　　　　　　**foreach** $Z \in Reached \setminus \{X\}$ **do**
　　　　　　　　$mw \leftarrow min(w(XY), minWeight(X, Z))$;
　　　　　　　　$minWeight(Y, Z) \leftarrow mw; minWeight(Z, Y) \leftarrow mw$;
　　　　　　　　**if** $mw = w(YZ)$ **then**
　　　　　　　　　　Add $YZ$ to $E'$;
　　　　　　　　**end**
　　　　　　**end**
　　　　　　Add $Y$ to $Reached$ and to $Queue$;
　　　　**end**
　　**end**
**end**
return $(\mathcal{E}, E')$;

---

**Theorem 7.** *Given a join tree and the weighted line graph of an $\alpha$-acyclic hypergraph $H$, the algorithm UJ-min-weight computes the union join tree of $H$ in $O(p^2)$ time.*

**Proof.** Correctness follows from the fact that by Characterization 13 the following proposition $P$ is clearly an invariant of the main for each and while loops.

　　$P$: $\forall \{X, Y\} \subseteq \mathcal{E}$, if $\{X, Y\} \subseteq Reached$, then ($minWeight(X, Y)$ is the minimum weight of an edge of $P_T(X, Y) \wedge (XY \in E' \Leftrightarrow XY$ is an edge of $UJ(H)))$; otherwise, ($XY \in E' \Leftrightarrow XY \in E_T$).

　　The algorithm runs in $O(p^2)$ time by numbering the elements of $\mathcal{E}$ from 1 to $p$ and storing the values of $MinCard(X, Y)$ for each $(X, Y)$ in $\mathcal{E}^2$, such that $X \ne Y$ in an array $p \times p$. □

　　By Characterization 11, the complexity bounds in function of $n$, $m$, and $\overline{m^+}$ presented in Section 4 extend to each connected $\alpha$-acyclic clutter $H$ replacing $\overline{m^+}$ by $\overline{m}$, as the graph $G = 2SEC(H)$ is equal to $G^+$. In fact, they also hold for each $\alpha$-acyclic clutter, replacing $m$ by $n + m$. This follows from the fact that the bounds of the parameters $p$, $s$, and $s_\triangle(T)$ by functions of $n$, $m$, and $\overline{m^+}$ extend to $\alpha$-acyclic clutters.

**Property 12.** *For each $\alpha$-acyclic clutter $H$, $p \le n$, $s \le n + m$, and for each join tree $T$ of $H$, $s_\triangle(T) \le n + \overline{m}$.*

**Proof.** By Characterization 11, these inequalities hold if $H$ is connected. It can be proved that they also hold if $H$ is disconnected by checking that the proofs of these inequalities given in Section 4 still hold. It can also be directly checked as follows. Let $H_1, \dots, H_k$ the

connected components of $H$, and for each $i$ in $[1, k]$ and each variable $v$, let $v_i$ be the value of $v$ in $H_i$. Then, $p = \Sigma_{i=1}^{k} p_i \leq \Sigma_{i=1}^{k} n_i = n$. Similarly, $s \leq n + m$. For $s_{\triangle}(T)$, we have $s_{\triangle}(T) = \Sigma_{i=1}^{k} s_{\triangle}(T_i) + nb_1$, where $nb_1 = \Sigma_{XY \in E_T, X \cap Y = \varnothing} |X| * |Y|$ and $\overline{m} = \Sigma_{i=1}^{k} \overline{m_i} + nb_2$, where $nb_2 = \Sigma_{\{i,j\} \subseteq [1,k]} |V_i| * |V_j|$. As $nb_1 \leq nb_2$, it follows that $s_{\triangle}(T) \leq n + \overline{m}$. $\quad\square$

**Corollary 3.** *The complexity bounds in function of $n$, $m$ and $\overline{m^+}$ presented in Section 4 hold for each $\alpha$-acyclic clutter $H$, replacing $m$ by $n + m$ and $\overline{m^+}$ by $\overline{m}$.*

If $H$ is an $\alpha$-acyclic hypergraph, which is not a clutter, the values of $p$, $n$, and $s_{\triangle}(T)$ may be exponential in $n$. It is the case of the hypergraph $H = (V, P(V) \setminus \{\varnothing\})$, which is $\alpha$-acyclic, since $V$ is a hyperedge of $H$ (the tree whose edges are the pairs of hyperedges containing $V$ is a join tree of $H$).

## 7. Conclusions

In this paper, we provide two efficient algorithms to compute the atom graph of a graph and extend them to compute the union join graph of an $\alpha$-acyclic hypergraph.

Our algorithms, in the general case, compute the atom graph at no extra cost than computing the atoms.

It would be interesting to explore the class of graphs which are isomorphic to atom graphs and to provide a recognition algorithm for this class.

**Author Contributions:** A.B. and G.S. authors contributed to each part of the work. All authors have read and agreed to the published version of the manuscript.

**Funding:** This research received no external funding.

**Conflicts of Interest:** The authors declare no conflict of interest.

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
