# Peer review of "Computing the Atom Graph of a Graph and the Union Join Graph of a Hypergraph"

_algorithms, doi:10.3390/a14120347_

Round 1

Reviewer 1 Report

  • In the abstract part, what does the notion $n$ mean?
  • In the abstract part, it seems that the time complexity of your method, n^(alpha)logn, is larger than n^(alpha).
  • Figure 1 needs more explanation. 
  • It will be better if you can show some speedup results on real-world graphs.

Author Response

  • In the abstract part, what does the notion $n$ mean?

$n$ is the order if the  graph, i.e. the number of its vertices. We have added this precision, though this notation is well-known in the community

  • In the abstract part, it seems that the time complexity of your method, n^(alpha)logn, is larger than n^(alpha).

Sure !

  • Figure 1 needs more explanation. 

You are right, some explanations have been added.

  • It will be better if you can show some speedup results on real-world graphs.

Sure, but unfortunately, we cannot. We have only worked from a theorical point of view. If this paper is published, may be some reseachers working on applications will be able to provide these useful informations which will complete our work.

Reviewer 2 Report

  1. The topics approached in the paper have a high degree of difficulty.
  2. The paper introduces two efficient algorithms for computing the atom graph of a graph. Also, the introduction of the union join graph of an α-acyclic hypergraph is made with a strong argumentation on this concept.
  3. The determination of the union join graph of an α-acyclic hypergraph is based on the observation that the atoms of a graph G = (V, E) can be seen as the hyper-edges of an α-acyclic hypergraph.
  4. The authors extend the results for α-acyclic hypergraphs: a) computation of the union join graph of an α-acyclic hypergraph; b) for the general case, the algorithms compute the atom graph without supplementary costs.
  5. The algorithms are applied for atomic graphs, to compute in an efficient manner the union join graph.
  6. The results presented in the paper are important both from the theoretical and from the practical point of view.
  7. Please offer several contexts where the new results can be applied. This would bring more value to the paper.

Author Response

1.The topics approached in the paper have a high degree of difficulty.

2.The paper introduces two efficient algorithms for computing the atom graph of a graph. Also, the introduction of the union join graph of an α-acyclic hypergraph is made with a strong argumentation on this concept.

3.The determination of the union join graph of an α-acyclic hypergraph is based on the observation that the atoms of a graph G = (V, E) can be seen as the hyper-edges of an α-acyclic hypergraph.

4.The authors extend the results for α-acyclic hypergraphs: a) computation of the union join graph of an α-acyclic hypergraph; b) for the general case, the algorithms compute the atom graph without supplementary costs.

5.The algorithms are applied for atomic graphs, to compute in an efficient manner the union join graph.

6.The results presented in the paper are important both from the theoretical and from the practical point of view.

7.Please offer several contexts where the new results can be applied. This would bring more value to the paper.

Thank you very much for all these positive comments.

Concerning item 7., we are aware that offering several contexts where the new results can be applied would bring more value to the paper. Unfortunately, we cannot do it. We have only worked from a theorical point of view. If this paper is published, may be some reseachers working on applications will be able to provide these useful informations which will complete our work.

Reviewer 3 Report

This paper presents efficient algorithms that computes the atom graph of a given graph G, i.e., a graph whose vertices represent, in some sense, clique minimal separators of G and whose edges represent intersections between clique minimal separators. The time complexity of the algorithms depend on the matrix multiplication exponent. The authors extend their results to \alpha-acyclic hypergraphs. 

Given the short time provided to review the paper, it is not possible to check all the details carefully. Therefore, I could not check the correctness of all the proofs, although I am confident that they are correct. The results contained in this paper are interesting enough and suitable to be published.

The paper is also in a good shape, but I recomment the authors to add the main ideas because reading a list of definitions, lemmas, characterizations, and properties, without providing high-level ideas, is boring and does not help the reader to understand the paper. Especially in journal submissions the authors should make some efforts to guide the reader through the paper.

Here are some minor comments:

- The almost standard notation used to denote the matrix multiplication exponent is \omega. So you should substitute \alpha by \omega. You should also specify both in the abstract and in the introduction, what is the current best-known upper bound on the value of \omega.

- You should provide a high-level description of \alpha-acyclic hypergraphs in the abstract.

- You should specify if G_1 \cup G_2 is either a graphs or a hypergraph. In the preliminaries you do not say explicitly whether G_1 \cup G_2 is a graph or a hypergraph.

- N_G(C) is undefined.

- I cannot understand the sentence "S is a minimal separator if S has at least 2 full components, and S ...". I think this sentence is wrong as it does not match the definitions of minimal separator and minimal ab-separator.

- You should define what a chordless path is. Moreover, I cannot understand the second part of the definition of 2-pair, where you claim "or equivalently such that N(x) \cap N(y) is a minimal xy-separator of G". This equivalent definition is not straightforward and you should either prove it or provide a reference to a paper/book containing the proof.

- Why is the number of 2-pairs upper bounded by \bar m?

Author Response

This paper presents efficient algorithms that computes the atom graph of a given graph G, i.e., a graph whose vertices represent, in some sense, clique minimal separators of G and whose edges represent intersections between clique minimal separators. The time complexity of the algorithms depend on the matrix multiplication exponent. The authors extend their results to \alpha-acyclic hypergraphs. 

Given the short time provided to review the paper, it is not possible to check all the details carefully. Therefore, I could not check the correctness of all the proofs, although I am confident that they are correct. The results contained in this paper are interesting enough and suitable to be published.

The paper is also in a good shape, but I recomment the authors to add the main ideas because reading a list of definitions, lemmas, characterizations, and properties, without providing high-level ideas, is boring and does not help the reader to understand the paper. Especially in journal submissions the authors should make some efforts to guide the reader through the paper.

 *

We indeed made efforts to provide high-level ideas, but in the same time, we tried to make the paper as self-sufficient as possible, which implies that a great number of definitions have to be recalled. They are certainly very boring and difficult to read for a non-specialist. The pape ris aimed to be read by a reader who has some knowledge about minimal separators : he will jump over most of the numerous definitions and will rapidly reach the importnt part of the paper, which is different algorithms computing the atom graph of a graph. The reviewer is not familiar with the notion of minimal separators since his definition of the atm graph is erroneous. Its vertices do not represent the clique minimal separators but are the atoms, which are the maximal vertex sets inducing subgraphs having no clique separator. Its edges are the pairs of atoms whose intersection is a clique minimal separator. Thus the edges, and not the vertices, represent, in some sense, clique minimal separators of G. This is not a reproach, but just a fact that the revewer is not familiar with these notions, which explains that he may have some difficulty to read the paper.

It is true that there are many Lemmas, Properties and Characterizations. Some of them recall known results, the others are new results. Most of t hem are used in the proofs of the algorithms to make them readable. We have tried to mke the proofs at the same time complete and rigorous on one hand, and as simple and readable as possible on the other hand, which was not easy work because the subject is hard.

**

Here are some minor comments:

- The almost standard notation used to denote the matrix multiplication exponent is \omega. So you should substitute \alpha by \omega. You should also specify both in the abstract and in the introduction, what is the current best-known upper bound on the value of \omega.

*

Done

**

- You should provide a high-level description of \alpha-acyclic hypergraphs in the abstract.

*

Done

**

- You should specify if G_1 \cup G_2 is either a graphs or a hypergraph. In the preliminaries you do not say explicitly whether G_1 \cup G_2 is a graph or a hypergraph.

*

Done

**

- N_G(C) is undefined.

*

Right, this definition has been added.

**

- I cannot understand the sentence "S is a minimal separator if S has at least 2 full components, and S ...". I think this sentence is wrong as it does not match the definitions of minimal separator and minimal ab-separator.

*

This sentence is not wrong. Note that it contains 2 characterizations and not 2 definitions. We replace « if » by if and only if » to make this clear. Here is a proof :

It is sufficient to prove that "S is a minimal ab-separator if and only if  a and b lie in 2 different full components of S.

Let Ca and Cb be the components of S containing a and b respectively.

We assume that S is a minimal ab-separator. As S is an ab-separator, Ca and Cb are distinct. By definition of Ca, N(Ca) is a subset of S and as Ca and Cb are distinct, N(Ca) is an ab-separator. As S is a minimal ab-separator and, N(Ca) is an ab-separator which is a subset of S, N(Ca) = S, so Ca is a full component of S. In the same way, Cb is too.

Conversely, we assume that Ca and Cb are 2 different full components of S. As Ca and Cb are distinct, S is an ab-separator. Let us show that it is a minimal one. Let S’ be a strict subset of S, let us show that S’ is not an ab-separator. Let x in S-S’. As x is in N(Ca) and N(Cb) there is a path betwween a and b containing x and whose other vertices are either in Ca or Cb. So S’ is not an ab-separator, and therefore S is a minimal ab-separator.

**

- You should define what a chordless path is.

*

Done

**

 Moreover, I cannot understand the second part of the definition of 2-pair, where you claim "or equivalently such that N(x) \cap N(y) is a minimal xy-separator of G". This equivalent definition is not straightforward and you should either prove it or provide a reference to a paper/book containing the proof.

*

Actually this equivalence is relatively straightforward. Here is a proof by equivalence :

The following propositions are equivalent :

Every chordless path between x and y is  of length 2

Every chordles path between x and y contains a vertex of N(x) \cap N(y)

Every path between x and y contains a vertex of N(x) \cap N(y)

N(x) \cap N(y) is an xy-separator

N(x) \cap N(y) is a minimal  xy-separator (since in the general case, it is a subset of any xy-separator)

We believe that this proof (or a similar one) appears somewhere in the literature in some old paper or book, but we do not know where.

.**

- Why is the number of 2-pairs upper bounded by \bar m?

*

A 2-pair is a pair of non-adjacent vertices, i.e. an edge of \bar G. It follows that their number is upper bounded by the number of edges of \bar G, which is \bar m.

**

Round 2

Reviewer 3 Report

The authors addressed all the comments. From my point of view, the paper is ready to be published.